# Experimental and Computational Study of Optimized Gas Diffusion Layer for Polymer Electrolyte Membrane Electrolyzer

**DOI:** 10.3390/ma16134554

**Published:** 2023-06-23

**Authors:** Javid Hussain, Dae-Kyeom Kim, Sangmin Park, Muhammad Waqas Khalid, Sayed-Sajid Hussain, Ammad Ali, Bin Lee, Myungsuk Song, Taek-Soo Kim

**Affiliations:** 1Industrial Technology, University of Science and Technology, Daejeon 34113, Republic of Korea; javidmohsin77@kitech.re.kr (J.H.); jhsm8920@kitech.re.kr (S.P.); waqas@kitech.re.kr (M.W.K.); ammad5125@kitech.re.kr (A.A.); leebin@khu.ac.kr (B.L.); 2Korea Institute for Rare Metals, Korea Institute of Industrial Technology, Incheon 21999, Republic of Korea; kyeom@kitech.re.kr; 3Chemical Engineering and Applied Chemistry, Chungnam National University, Daejeon 34134, Republic of Korea; sayedsajidh506@gmail.com

**Keywords:** PEMFC, GDL, tape casting, GeoDict, optimization, physical properties

## Abstract

Polymer electrolyte membrane fuel cells (PEMFCs) and PEM electrolyzer are emerging technologies that produce energy with zero carbon emissions. However, the commercial feasibility of these technologies mostly relies on their efficiency, which is determined by individual parts, including the gas diffusion layer (GDL). GDL transfers fluid and charges while protecting other components form flooding and corrosion. As there is a very limited attention toward the simulation work, in this work, a novel approach was utilized that combines simulation and experimental techniques to optimize the sintering temperature of GDL. Ti_64_ GDL was produced through tape casting, a commercial method famous for producing precise thickness, uniform, and high-quality films and parameters such as slurry composition and rheology, casting parameters, drying, and debinding were optimized. The porosity and mechanical properties of the samples were tested experimentally at various sintering temperatures. The experimental results were compared with the simulated results achieved from the GeoDict simulation tool, showing around 96% accuracy, indicating that employing GeoDict to optimize the properties of Ti_64_ GDL produced via tape casting is a critical step towards the commercial feasibility of PEMFCs and electrolyzer. These findings significantly contribute to the development of sustainable energy solutions.

## 1. Introduction

Global warming, the high emission of carbon gases, and the energy crisis compel the world to move toward green energy. The fuel cell is one of the crucial technologies to solve this problem. The fuel cell is the best alternative for storing energy rather than batteries as it has an issue of self-discharge. There are many applications of fuel cells such as portable electronic devices, vehicles, power generation, transportation, backup power system, aerospace, military, and marine [1]. The basic five types of fuel cells on the basis of electrolytes are (PEM) fuel cells, solid oxide fuel cells, alkaline fuel cells, molten carbonate fuel cells, and phosphoric acid fuel cells. Among them, the PEM fuel cell received more attention due to the quick start and shut down, high efficiency, low operating temperature, and simple design. The PEM fuel cell uses hydrogen as the intake fuel and converts it into electrical energy [2,3]. The high combustion value, purity, and stability of hydrogen make it one of the most attractive sources of renewable energy [4]. To sufficiently provide hydrogen to the PEM fuel cell, enough hydrogen production is needed [5].

There are various techniques through which hydrogen can be produced. The most advanced technique is the alkaline electrolyzer, but its drawbacks include a corrosive electrolyte, a low maximum current density, and low operating pressure. In addition, the solid oxide electrolyzer has excellent efficiency and quick reaction time, but the high-temperature operation is the main limitation. Hence, the production of hydrogen by water electrolysis using a PEM electrolyzer can be a prominent solution. The PEM electrolyzer is at the door of commercialization. This electrolyzer allows various advantages over other electrolyzers such as the simple design, high productivity, fast response system, compactness, being safe at high current densities, easy maintenance, and environmental friendliness [6,7,8,9].

PEM electrolyzer convert electrical energy into chemical energy with oxygen as a by-product. It performs electrolysis of water into oxygen and hydrogen. The proton travels through the proton exchange membrane which is a solid electrolyte, while the electron travels through the external circuit, reacts again at the cathode side and releases hydrogen as the final product. The excess water and oxygen are the by-product in the whole process, which is very important for the control of the greenhouse effect [10,11].

The cathode and anode reactions of the electrolyzer are given as
Anode: 2H_2_O ↔ O_2_ + 4H^+^ + 4e
Cathode: 4H^+^ + 4e^−^ ↔ 2H

The overall reaction is as [7]
2H_2_O + electrical energy ↔ 2H_2_ + O_2_

The electrolyzer is composed of the PEM membrane, anode, cathode, GDL, bipolar plate, and end plate. The major challenge in the electrolyzer is the cost of its individual parts. Since the output of one single cell is very limited, multiple cells are connected in series to increase the final production of hydrogen. The efficiency of the whole cell is based on the performance of each component including GDL [8].

### 1.1. GDL

The gas diffusion layer (GDL) is one of the essential parts of PEMFC and electrolyzer. It greatly affects the performance of PEMFC and facilitates fluid movement to and from the electrodes. GDL also shields other components from corrosion and erosion during the process. GDL, which exists in single and double layer configurations and is comprised of porous material, has a number of industrial, biomedical, and power generation uses. GDL must be designed with high thermal and electrical conductivity, strong mechanical properties, high corrosion resistance, and high permeability. Good thermal and electrical properties are important for efficient ion transport and fluid management. The high mechanical strength along with the high porosity is required to withstand the fluid pressure at high temperatures and ensure the high permeability of the fluid [12]. In addition, the hydrophilic and hydrophobic properties must also be considered in the design. The hydrophobic nature of certain materials can help manage water in and out of fuel cells, and coatings may be necessary for this purpose. To ensure all these properties, proper design and open pores are necessary for accurate handling of capillary pressure, and uniform transportation in order to avoid transport and flooding problems. The porosity and pore morphology of a GDL significantly affect its performance and, consequently, the efficiency of a PEM fuel cell. To achieve all these properties, it is important to select the appropriate material for the GDL [13,14].

Conventionally, carbon and steel were used as GDL materials, but their commercial feasibility was limited due to their mechanical and chemical degradation. Steel has high mechanical strength, but corrosion limits the commercialization of steel. The strong corrosion resistance of GDL is essential in the corrosive environment of the fuel cell and electrolyzer due to the presence of high reactive ions of H_2_ and O_2_ [15]. To solve this problem, titanium and its alloy can be a good candidate due to it having the same strength as steel with almost half density and high corrosion resistance. The high strength and light weight of titanium are also important for the smart manufacturing and weight control of the fuel cell and electrolyzer stack. Minh Young et al. reported that titanium is ten times more resistant to degradation compared to carbon [16].

This makes titanium a potentially attractive material for use in GDL and other applications where degradation resistance is important. The platinum coating makes this material very much feasible for commercial use with satisfactory corrosion resistance and mechanical strength [6,17]. After the selection of appropriate material, the fabrication and optimization of various properties of GDL is very important and challenging. Experimental optimization is very difficult due to the very low thickness of GDL and its porous nature. There is very limited attention given to the simulation work; therefore, in this study, GeoDict was employed for the modeling, simulation, and optimization process.

### 1.2. GeoDict

Currently, there is limited simulated research on titanium GDL manufactured by powder metallurgy. However, software prediction is expected to play a crucial role in the future for GDL design and property optimization. Computational fluid dynamics (CFD) provides insights into macroscopic fluid behavior, while finite element analysis (FEA) focuses only on mechanical aspects. Molecular dynamics (MD) simulations has specialty in offering atomistic-level insights. GeoDict possesses distinctive functionalities that enable the generation of realistic 3D micro-sized geometries, simulation of fluid flow and transport phenomena, and analysis of the mechanical behavior of GDL materials. These exclusive capabilities make GeoDict an ideal tool for investigating complex transport phenomena and optimizing the performance of GDLs in PEMFCs.

The GeoDict simulation tool (Release version: service pack 4) is a very adaptable and flexible software that needs several inputs based on real calculations, such as powder size and prototype dimensions, the volumetric porosity, distribution of the powder, volumetric shrinkage, etc. This also includes some manual addition of physical characteristics of the material. For instance, in this article, Ti_64_ was used and various modules were employed for the simulation with the manual inputs of the basic properties. The young modulus was measured and a comparison was then made with the experimental data. Moreover, several predictions were made to analyze and optimize other properties of GDL at various sintering temperatures (porosity).

Two studies, one by Gervais et al. and the other by Sudhakar et al., showed that GeoDict is highly accurate. Gervais et al. were able to achieve 97% accuracy, while Sudhakar et al. found that there was only a 0.07 deviation (0.3%) when comparing simulated water permeability with experimental results. This shows the verification and the authentication of GeoDict. Overall, the GeoDict simulation tool is widely recognized as a reliable and precise tool for simulating and predicting various characteristics of porous materials [18,19]. In a study conducted by Dennis Hoch et al., a comparison was made between GeoDict and StarCCM+, using the experimental results. The findings revealed that while StarCCM+ exhibited almost 1% higher accuracy, its simulation time was significantly longer and less convenient compared to GeoDict [20]. According to M. Amin et al. [21], who compared the results of GEODICT and COMSOL with the experimental results, GeoDict demonstrated a high level of accuracy in its predictions. H. Bai et al. utilized FilterDict to simulate the effective diffusion coefficient and filtration performance in relation to particle diameter. The results showed a deviation of approximately 0.6% from the experimental data, thus confirming the remarkable accuracy of GeoDict [22].

In conclusion, this paper aimed to investigate and develop a single-layer Ti_64_ GDL via tape casting. The tape was cast and sintered at different temperatures that ranged from 800 to 1400 °C. Then, the characterization was carried out and nanoindentation was used to find the mechanical properties and few readings were taken of each sample and the average was then carried out to minimize the error. For the validation of the GeoDict, first, a comparison was made of mechanical properties between the experimental and simulated data. A very decent agreement was seen. On the basis of these simulations, a further prediction and optimization of sintering w.r.t porosity was performed to successfully fabricate and develop a Ti_64_ GDL with high efficiency.

## 2. Experiment and Modeling

### 2.1. Material and Fabrication

Spherical APATM Ti-6Al-4V grade 23 powder of (10–60) μm size manufactured by AP& C a GE Additive company was taken as the starting material, as shown in Figure 1b. The powder size was measured with the particle analyzer. An aqueous-based slurry was preferred to avoid the hazardous effect. The slurry consisted of the binder polyvinyl alcohol (363073-500G, Sigma-Aldrich, USA) and the dispersant ammonium polymethacrylate (DARVAN^®^ C-N, R.T. Vanderbilt, USA). To further enhance the rheological properties, the plasticizer polyethylene glycol (300, Sigma-Aldrich, Germany) was added to the slurry. Additionally, the surfactant Tergitol TMTMN-6 was utilized to improve the surface properties of the powder. The composition of each additive was carefully optimized through the use of a viscometer. The optimized composition of each component w.r.t to the powder is shown in Table 1.

The mixing process was performed using a centrifugal mixer, with a one-step mixing technique at 1000 rpm for 10 min. Subsequently, de-foaming was carried out for 40 min at 600 rpm to remove any trapped air bubbles. The slurry was poured into a doctor blade machine, where the thickness was precisely controlled using the doctor blade. The initial thickness was set at 500 μm and it was almost 300–350 μm after drying. The tape was cast on the doctor blade machine at 1.5 mm/s speed and 18.5 °C temperature. The slow speed and low temperature were maintained to avoid cracking of the tape. After drying for 24 h in room temperature, the green tape was cut into a 3 mm × 3 mm pieces and debinding was carried out at 1 °C/min. Finally, the sintering was conducted at a heating rate of 5 °C/min, with a holding time of 2 h at a temperature range of 800 °C to 1400 °C. The whole process of tape casting is shown in Figure 1a.

The sintering and debinding were performed in a high temperature alumina furnace with high purity Ar atmosphere to prevent oxidation and other contamination. The porosity of each sample were measured by the image analyzer software version 1.0. This sophisticated instrument employs powerful algorithms to analyze the intricate microstructures and extract crucial porosity information. By employing high-resolution imaging and precise image segmentation techniques, we gained a comprehensive understanding of the sample’s porosity. This meticulous approach ensured precise and reliable porosity measurements, enabling us to delve deeper into the material’s properties and gain valuable insights. The samples were analyzed using scanning electron microscopy (SEM) and X-ray diffraction (XRD) techniques. Additionally, nano-indentation was employed to measure the shear modulus. The pressure and time for nano-indentation were set at 100 mN and 60 s, respectively. Several points were selected at different position and then average were taken to limit the error. For the modeling and simulation, GeoDict software tool was employed.

### 2.2. Modeling

#### Establishment of Grain Models

GeoDict is a user-friendly and integrated software, developed by Math2Market for multi-scale 3D image processing, material modeling, visualization, and property analysis [23]. The grain geo sub-model was used for the 3D modeling of spherical particles that were used for the fabrication of GDL in tape casting. GDL modeling was started with the dimension of 300 × 300 × 300 microns. The iteration and random seed were set to 1 and 45, respectively. During the modeling process, the air was selected as a fluid in the pores. The density was given w.r.t each sintering temperature that was measured with the image analyzer. The Gaussian powder distribution was selected with the size range of 5–65 microns, as measured by the powder size analyzer. The pile grain model was employed to construct a new model based on the real experimental inputs. After that, a distribution process was carried out and 140 iterations were selected for a more uniform distribution. The higher iteration resulted in more uniform distribution. The shift distance/voxel of 10 were used with a random seed of 1. The sintering process was performed based on the porosity of each sample. The sintering procedure was carried out with precision, considering the unique porosity of each sample. The whole process of 3D modeling, including some 3D visualization, is shown stepwise in Figure 2. The 3D models before and after distribution is shown in Figure 2a,b.

After distribution, the sintering was carried out, which depended on the shrinkage percentage as shown in Figure 2c. To gain a deeper understanding of the process, visual representations of FlowDict, ElastoDict, PoroDict, and ConductoDict can be seen in Figure 2d–g. This illustration highlights the systematic and meticulous approach that was taken to ensure the highest degree of accuracy and quality in the results.

### 2.3. Simulation and Prediction

The modeling process included generating a 3D model, distributing it, and then, sintering it. However, simulation was performed using various models, as shown in Figure 2, to predict the results at various porosity, representing the sintered sample at each temperature.

#### 2.3.1. ElastoDict

The ElastoDict model was used to simulate the mechanical properties of the sample. Since some values of the titanium alloy (Ti_64_) were not in the list of the material of GeoDict, the elastic modulus, shear modulus, and poison ratio of the titanium were given manually, which were 114 Gpa, 44 Gpa, and 0.33, respectively [24,25,26]. The pores and fluid were assigned the values 0 for each parameter. Based on these values, the mechanical properties at various porosity were simulated. These results were compared with the experimental values and further predictions were made to achieve optimal porosity and sintering conditions.

#### 2.3.2. FlowDict

Permeability is a crucial aspect of GDL and plays a significant role in its performance. The FlowDict was used to simulate the effect of porosity on permeability and velocity in a porous material (GDL). The fluid flow in the porous material (GDL) usually takes place at low Reynolds number, which can be explained by the incompressible stokes equation and Darcy’s law as given
$$\vec{u}=\frac{K}{µ}\left(\nabla p-\vec{f}\right)$$

In this equation, $\vec{u}$ is the fluid velocity, ∇p is the pressure, K is the permeability, $\vec{f}$ is the force density and, µ is the fluid velocity. The Stokes–Brinkman equation was chosen due to the low velocity of the fluid in the porous material (GDL). The flow permeability tensor found from Darcy’s law was used to find the permeability tensor
$${\vec{v}}^{i}=-{µ}^{-1}\;\left[\begin{array}{ccc} Kxx & Kxy & Kxz\\ Kyx & Kyy & Kyz\\ Kzx & Kzy & Kzz \end{array}\right]\;\nabla p^{i}$$
where $\vec{v}$ is the average velocity, while i = 1, i = 2, and i = 3 represent the pressure drop in the X, Y, and Z direction, respectively. The three input parameters required for the flow experiment are the structure of porous material, the fluid, and the process parameters such as mass and flow rate, etc. [27,28]. The flow in the porous material GDL is very small; hence, the Stokes–Brinkman equation highly suits this phenomenon.
$$\nabla p=-µK^{-1}\;\vec{v}+\nabla \cdot {µ}^{*}\;(\nabla \vec{v}+\nabla {\vec{v}}^{T})$$
$$\nabla \cdot \vec{v}=0$$
where p is the fluid pressure, $\vec{v}$ is the velocity, K is the permeability tensor, µ and µ* are the fluid viscosity, and the fluid effective viscosity, respectively. For the prediction of permeability, water was taken as the fluid. Density and dynamic viscosity as an inputs were 998 kg/m^3^ and 1.787 × 10^−3^ kg m^−1^ s^−1^, respectively [29].

#### 2.3.3. ConductoDict

The thermal properties are also a crucial feature of the GDL, which keeps the cell cool and also helps in the removal of fluid from the cell and makes it easy in the vapor form. Additionally, the electrical conductivity affects ion transport; hence, a high electrical conductivity is desirable for the GDL. ConductoDict was employed for the simulation and prediction of these two properties. The thermal conductivity is based on Fourier’s law
$$\dot{q}\;=\;-K\nabla T$$
where K is the thermal conductivity, $\dot{q}$ is the heat flux, and T is the temperature. As the thermal conductivity is the second-order tensor, hence
$$K=\left(\begin{array}{ccc} K11 & K12 & K13\\ K21 & K22 & K23\\ K31 & K32 & K33 \end{array}\right)$$

The material is isotropic and, therefore, instead of K, the scalar constant can be applied.

For electrical conductivity, Ohm’s law is applied
j = −σ ∇ φ
where j is the current density and φ is the electric potential [30].

Similar to thermal conductivity, the electrical conductivity is also the same in all direction; hence, the alpha is treated as a scalar.

Since the GeoDict did not contain all of the information of titanium alloys, the thermal and electrical conductivity of Ti_64_ and water (flows in GDL) were manually added. The thermal and electrical conductivity of Ti_64_ were 561,798 S/m and 6.7 W/mk, respectively. The thermal conductivity of the fluid (GDL) moving through the porous material was 1.787 × 10^−3^ W/mk, while electrical conductivity was 0.0005 S/m [31,32].

#### 2.3.4. PoroDict

The structure of the pores significantly impacts all of the porous material’s properties. The PoroDict was used to thoroughly examine the pores in the GDL at each porosity. The differential pore distribution in the PoroDict is based on the equation below.
Pore volume distribution = V_cum_ (d_i+1_) − V_cum_ (d_i_)/(ln (d_i+1_) − ln (d_i_)).m
where d is the pore diameter, v is the volume fraction, V_cum_ is the cumulative volume fraction, and m is the mass. The mass is computed on the base of the densities of the material. In addition, the bubble point pc was computed on the bases of the young Laplace equation
$$r=\frac{2\sigma }{pc}\cos\alpha$$
where α is the wetting phase contact angle, σ (sigma) is the surface tension, and r is through pore radius [33].

## 3. Result and Discussion

### 3.1. Model Verification and Analysis

The SEM microstructure of the gas diffusion layer (GDL) sintered at different temperatures was analyzed to better understand the properties of the material. As shown in Figure 3, at 800 °C, the particles of the GDL were not connected, leading to a brittle material. However, as the temperature increased to 900 °C, some particle connections formed and the porosity decreased. At 1000 °C, necking started to appear and continued to increase as the temperature was raised to 1100 °C and 1200 °C. When the sample was sintered at 1300 °C, the particles came together more closely, filling in the gaps, some grain formation began, which indicates a decrease in porosity and at 1400 °C, a completely sintered body with limited porosity and complete grain formation was achieved, providing optimal mechanical stability to the GDL. The 2D models generated by GeoDict at each porosity (sintering temperature) showed the same trend in microstructure, as seen in Figure 4. The comparison between the GeoDict-generated 2D structure and the SEM analysis showed that the model was generated using real experimental inputs and was a reliable representation of the material’s properties. Furthermore, the XRD data were analyzed to gain a comprehensive understanding of the sample composition. The XRD pattern of GDL samples sintered at various temperatures (800 °C, 900 °C, 1000 °C, 1200 °C, and 1400 °C) is shown in Figure 5.

As there were no specific PDF cards for the Ti_64_ alloy, two cards, 00-044-1294 and 00-009-0098, were matched to identify the phase and structure of the XRD peaks. After the analysis of peaks at various 2θ, it was revealed that the majority of the peaks corresponded to the alpha (hcp) phase of titanium [34,35]. However, a small amount of the beta (bcc) phase was also detected at a specific 2θ value of 40. This suggests that the material had a dual-phase microstructure, with a mixture of hcp and bcc phases coexisting. It is worth mentioning that depending on the thermomechanical processing of the material, the proportion of each phase can be altered; hence, this information is important to understand the microstructure–property relationship of the material. Additionally, the XRD pattern showed that binder and other organic compounds were completely removed.

After the microstructural analysis, the porosity at each temperature was determined, as shown in Figure 6a. The porosity was around 47% at the start of 800 °C and it decreased to 5% at 1400 °C. The lower porosity was preferred for the higher mechanical, thermal, and electrical properties, but as a consequence, the permeability and other flow properties decreased. For this reason, a deep study of these properties at each porosity is very important for better optimization of porosity and, indirectly, the sintering temperature.

### 3.2. Prediction of Various Properties via GeoDict

The mechanical properties of porous materials are critical to withstand the fluid pressure and to ensure their performance in applications such as fuel cells. The validation of the simulation software GeoDict was demonstrated by comparing the simulated and experimental young modulus, as shown in Figure 6b. The comparison showed good agreement, with an average error of around 4% as demonstrated in Figure 6c. In order to further study the effect of porosity on the mechanical properties, the shear modulus, bulk modulus, and lame modulus were predicted at each porosity, as represented in Figure 6d. The poison ratio was also analyzed in relation to porosity, as shown in Figure 7a. The results showed that the mechanical properties were highly sensitive to even small changes in porosity. The optimization of mechanical properties was crucial for ensuring the high performance of the gas diffusion layer (GDL) in fuel cells.

For further optimization, the relationship between porosity and permeability was simulated via GeoDict and the results are shown in Figure 7b. The permeability was in the range of 10^−12^ m^2^ up to 1000 °C, and it then decreased to 10^−13^ m^2^ and it became extremely less at higher temperature. This indicates that high sintering temperatures had a negative impact on the permeability. This trend was further confirmed by analyzing the average velocity and tortuosity, as shown in Figure 7b,c. The decrease in fluid velocity with decreasing porosity was a result of reduced available flow pathways, increased resistance to flow, enhanced fluid–solid interactions, and higher tortuosity within the porous material. These findings demonstrate the need for a balanced approach in optimizing the sintering temperature, taking into account both mechanical properties and flow properties to achieve optimal performance in the gas diffusion layer (GDL) of fuel cells. Furthermore, the relation of bubble point pressure and the porosity can be observed in Figure 7d. These results showed an easy bubble formation at higher porosity, which need to be further studied for the best control and optimization.

In addition to the mechanical and flow properties, the thermal and electrical properties of the GDL were also investigated. These properties play a crucial role in the transmission of charge and heat, which highly affect the overall efficiency of the cell. The relation of the porosity with the thermal and electrical conductivity can be seen in Figure 8a,b. With an increase in porosity, the electrical and thermal properties exhibited a discernible decline, highlighting a crucial relationship between these properties and the porosity of the material. This phenomenon can be attributed to several underlying reasons. Firstly, as porosity increased, the material’s effective cross-sectional area decreased. This reduction in the available pathway for electrical and thermal conduction led to a decrease in conductivity and thermal diffusivity. The presence of voids and air-filled spaces within the material interrupted the flow of electrons and heat, impeding their efficient transfer. Secondly, the interconnected voids and pores within the porous structure created additional interfaces, introducing resistance to the movement of charge carriers and thermal energy. These interfaces acted as barriers, hindering the smooth flow of electrons and heat through the material. Consequently, electrical and thermal conductivities were reduced. It is worth noting that the extent of the decrease in electrical and thermal properties depended on various factors, including the porosity volume fraction, pore size, and distribution. Materials with higher porosity and larger pore sizes tended to exhibit more pronounced decreases in these properties.

The morphology of the pores and grains within the GDL plays a crucial role in determining its overall properties. For example, the size, shape, and distribution of the pores decides the overall porosity and the flow properties of the material. Similarly, the shape, size, and orientation of the grains can have a significant effect on its mechanical, thermal and electrical properties. In summary, the porosity and grain morphology are essential microstructural features that significantly influence the macroscopic properties of the GDL. Some of the result related to the grains and pores at each sintering temperature (porosity) are as illustrated in Figure 9.

During the initial stages of the sintering process, the material exhibits a higher porosity, characterized by a low volume of grains and relatively large pore diameters, particularly open pores, as depicted in Figure 9a–d. As the temperature rises, the number of pores increased while their diameters decreased. Simultaneously, the number of grains decreased, and their volumes diminished with the escalating temperature. At higher porosity levels and larger pore diameters, a wider pathway was established for fluid flow, resulting in enhanced permeability and average velocity. These conditions facilitated a smoother and more defined movement of the fluid within the material On the other hand, at the high sintering temperature (lower porosity than 30%), the grain volume increased significantly while the pores volume declined. This shift from pores to grain ensured higher mechanical properties and lower flow properties. These understandings enable us to optimize the sintering temperature (porosity) to obtain the optimal mechanical, thermal, electrical, and flow properties.

### 3.3. Optimization of Sintering Temperature w.r.t Various Properties

In a world where we demand sustainable energy solutions, the optimization of PEM fuel cell and electrolyzer materials is crucial. We observed that mechanical, thermal, and electrical properties increased with the decrease in the porosity, but the permeability, tortuosity, and the average velocity decreased as the porosity declined. For this purpose, the higher and lower porosity was not suitable for the application of PEM fuel cell and electrolyzer.

The optimization of the sintering temperature for GDL fabrication in PEM electrolyzers is crucial, taking into account two key factors such as permeability and mechanical properties. Extensive studies showed that a permeability above 10^−12^ m^2^ is desirable for achieving high efficiency in GDLs. Unfortunately, the sample sintered at 1100 °C and above results in lower permeability values that did not meet this threshold, rendering them unsuitable for efficient applications [36,37,38,39,40,41].

On the other hand, lower sintering temperatures such as 800 °C and 900 °C may yield higher permeability, but they lack the necessary mechanical strength required for handling and testing in real stacks. These samples tend to be fragile and prone to breakage, making them impractical for implementation.

By selecting a sintering temperature of 1000 °C, a delicate balance was achieved between mechanical strength and permeability. Samples sintered at this temperature exhibited sufficient strength to withstand handling and fluid pressure, while still maintaining a relatively high permeability compared to high temperature (above 1000 °C) sintered samples. Overall, the optimization of the sintering temperature at 1000 °C ensures that the GDL exhibits both desirable mechanical strength and adequate permeability for efficient operation in PEM electrolyzers.

To achieve better results, there is a need for some special changes in the process to make, align, and control pores and grains. The pores and grains mainly depend on the sintering temperature, the heating and the cooling rate, the additive used in the slurry, the size and composition of powder, etc. In the bigger picture, these conditions control the overall properties of the whole material. However, this paper aimed to explore the optimization of only sintering temperature, and found that 1000 °C was the best optimal condition for the better performance. This study successfully optimized the sintering temperature and demonstrated that 1000 °C was the best sintering condition for the better performance of PEM fuel cell and electrolyzer materials.

## 4. Conclusions and Future Recommendation

In a world that is increasingly focused on sustainable energy solutions, the optimization of materials used in PEM fuel cells and electrolyzers is of utmost importance. The gas diffusion layer (GDL) is a critical component of PEM fuel cells, and its efficiency greatly impacts the overall performance of the system. This study demonstrated that tape casting is a highly effective method for producing Ti_64_ GDL. In addition, we observed that the flow properties increased with increasing po rosity, while mechanical properties, electrical, and thermal conductivity decreased. On the other hand, the sample sintered at higher temperatures had a reverse trend. The higher and lower temperature cannot be the ideal because at higher sintering temperature (porosity), the flow properties were very small and at lower sintering temperature, the material did not appreciate w.r.t to mechanical and conductive properties. Therefore, the optimized sintering temperature is 1000 °C and at this temperature, the mechanical, permeability, and the other properties showed an optimum value. Hence, we can say that the casted tape sintered at 1000 °C having porosity 36% is the optimized condition for this process. Moreover, the use of simulation tools such as GeoDict proved highly effective for optimizing the properties of Ti_64_ GDL produced via tape casting, and provided valuable insights for predicting and optimizing various properties.

The future of GDL fabrication lies in further increasing the porosity to enhance permeability without compromising mechanical properties. This can be achieved through the use of pore agents and high-temperature sintering techniques. Additionally, the application of fiber in the tape casting process can improve the properties of the Ti_64_ GDL. The control of pore structure is a crucial aspect of future research in GDL fabrication, and GeoDict simulation tool can provide a powerful information for the development of GDL with versatile properties. The coating of Ti_64_ for improved properties is an exciting avenue for future research. Overall, the optimization of GDL materials is a critical step towards the commercial feasibility of PEMFCs.

## Figures and Tables

**Figure 1 materials-16-04554-f001:**
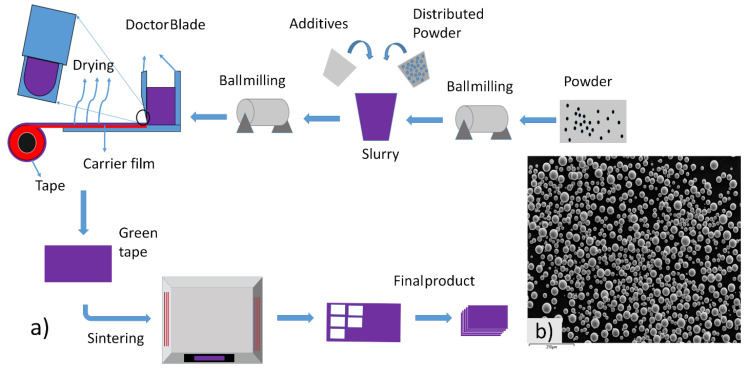
(**a**) Complete process of tape casting, and (**b**) SEM of Ti_64_ powder.

**Figure 2 materials-16-04554-f002:**
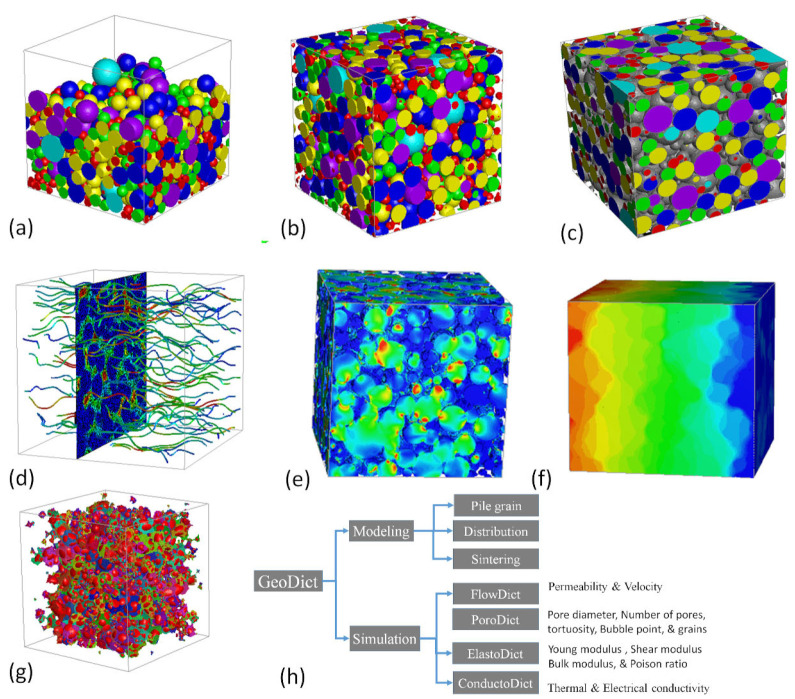
(**a**) 3D model before distribution, (**b**) distributed 3D model, (**c**) sintered 3D model, (**d**) fluid flow visual, (**e**) stress visual, (**f**) conductivity visuals, (**g**) pores visual, and (**h**) modeling and simulation of GDL by various module.

**Figure 3 materials-16-04554-f003:**
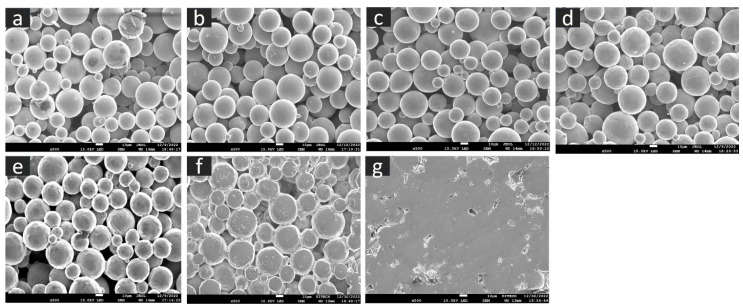
Microstructure of the GDL at various sintering temperature, (**a**) 800 °C, (**b**) 900 °C, (**c**) 1000 °C, (**d**) 1100 °C, (**e**) 1200 °C, (**f**) 1300 °C, and (**g**) 1400 °C.

**Figure 4 materials-16-04554-f004:**
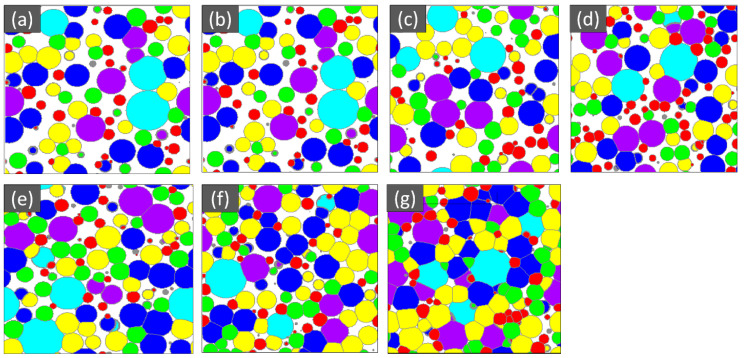
Microstructure of the GDL at various porosity (%), (**a**) 47, (**b**) 41, (**c**) 36, (**d**) 29, (**e**) 21, (**f**) 13, and (**g**) 5.

**Figure 5 materials-16-04554-f005:**
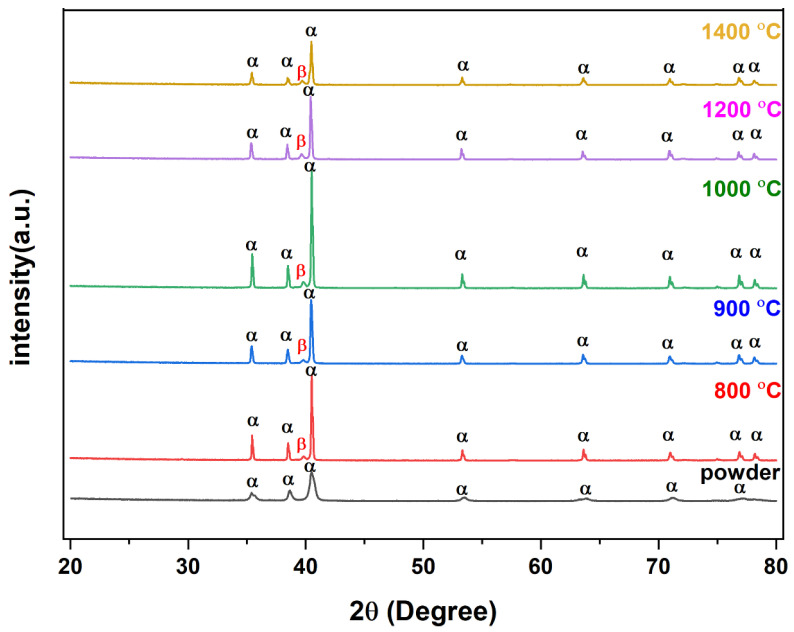
XRD analysis of powder and sintered Ti_64_.

**Figure 6 materials-16-04554-f006:**
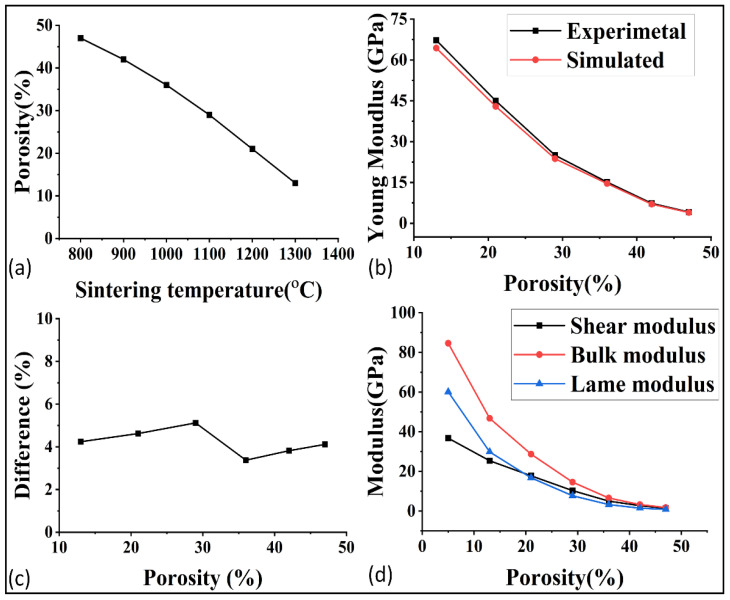
(**a**) Porosity of GDL at each sintering temperature, (**b**) experimental and simulated comparison of young modulus, (**c**) difference between the experimental and simulated result, and (**d**) relations of modulus with the porosity.

**Figure 7 materials-16-04554-f007:**
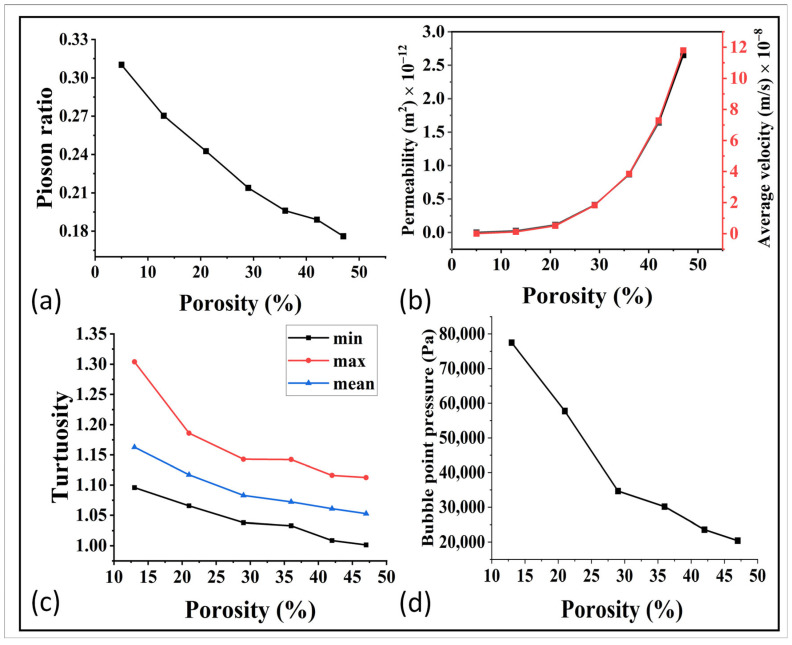
(**a**) Relation of porosity with the poison ratio. (**b**) Relation of Permeability and average velocity with porosity, (**c**) relation of tortuosity with porosity, and (**d**) porosity and bubble point relationship.

**Figure 8 materials-16-04554-f008:**
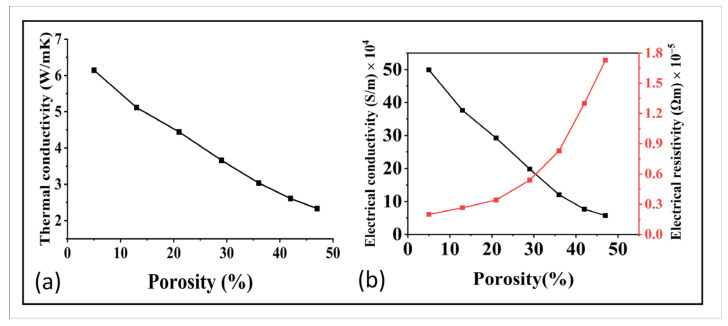
(**a**) Relation of porosity with thermal conductivity, and (**b**) relation of porosity and with electrical conductivity and electrical resistivity.

**Figure 9 materials-16-04554-f009:**
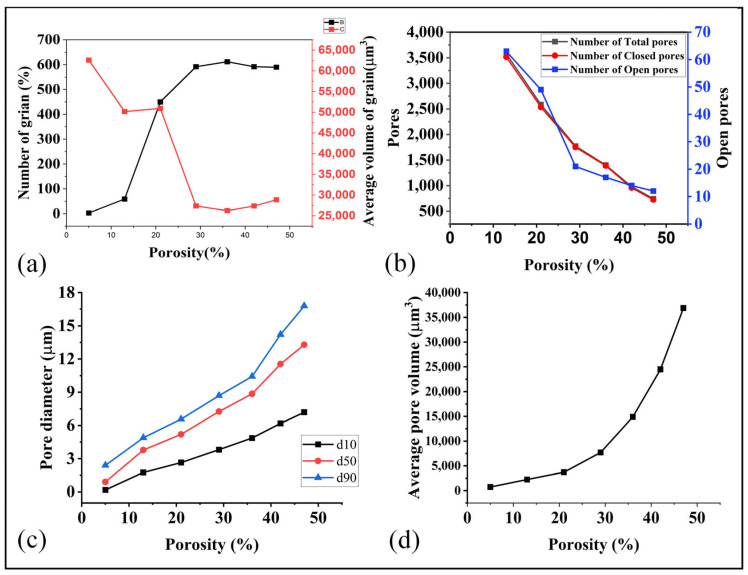
(**a**) Relation of porosity with number of grain and volume of grains, (**b**) porosity and pores, (**c**) porosity and pore diameter, and (**d**) porosity and average pore volume.

**Table 1 materials-16-04554-t001:** Composition of the additive w.r.t Ti_64_ powder.

Additive	Material	(Wt. %)
Water	Demi water	31.8
Dispersant	Ammonium Polymethacrylate	2
Binder	Polyvinyl alcohol	6.46
Plasticizer	Polymethacrylate	6.46
Surfactant	Tergitol TMN 6	2

## Data Availability

Not applicable.

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
