# Peer review of "Experimental and Computational Study of Optimized Gas Diffusion Layer for Polymer Electrolyte Membrane Electrolyzer"

_materials, 2023, doi:10.3390/ma16134554_

Round 1

Reviewer 1 Report

This article nicely describes the optimization of e Gas Diffusion Layer (GDL) materials for the technical and commercial feasibility of Polymer electrolyte membrane fuel cells (PEMFCs). The Gas GDL is a critical component of PEM fuel cells. The authors nicely present a series of comprehensive experimental for producing Ti64 GDL. The authors nicely proved that the control of pore structure is a crucial aspect of future research in GDL fabrication, and GeoDict simulation tool can provide a powerful information for the development of GDL 414 with versatile properties. I think this work is an immersive study in the field of PEMFCs. I think authors have presented the scientific data in an excellent manner. Therefore, I recommend publication in its present form. Though authors need to revise the whole manuscript e.g., crosscheck some spelling mistakes and grammatical errors through out the manuscripts before publication.

Though authors need to revise the whole manuscript e.g., crosscheck some spelling mistakes and grammatical errors through out the manuscripts before publication.

Author Response

Reviewer 1.

This article nicely describes the optimization of e Gas Diffusion Layer (GDL) materials for the technical and commercial feasibility of Polymer electrolyte membrane fuel cells (PEMFCs). The Gas GDL is a critical component of PEM fuel cells. The authors nicely present a series of comprehensive experimental for producing Ti64 GDL. The authors nicely proved that the control of pore structure is a crucial aspect of future research in GDL fabrication, and GeoDict simulation tool can provide a powerful information for the development of GDL 414 with versatile properties. I think this work is an immersive study in the field of PEMFCs. I think authors have presented the scientific data in an excellent manner. Therefore, I recommend publication in its present form. Though authors need to revise the whole manuscript e.g., crosscheck some spelling mistakes and grammatical errors throughout the manuscripts before publication.

Our response to the comment:

Thank you for your valuable feedback on our article. We appreciate your positive comments and we are pleased to hear that you found our work on producing Ti64 GDL to be comprehensive and well-presented. We agree with your observation that controlling the pore structure is a crucial aspect for future research in GDL fabrication, and we are glad that our study highlights the importance of this aspect. We are also delighted that you recognized the power of the GeoDict simulation tool in providing valuable information for the development of GDLs with versatile properties.

We appreciate your recommendation for publication in its present form. We take your suggestion seriously and will thoroughly revise the manuscript to address any spelling mistakes and grammatical errors before final publication. Your feedback will help us improve the overall quality of our work.

Once again, we would like to express our gratitude for your positive evaluation and constructive comments.

Reviewer 2 Report

The authors created a gas diffusion layer (GDL) by tape casting and measured the effects of sintering temperature on mechanical properties and porosity. Model GDLs corresponding to various sintering temperatures were created by piling up grains. It was confirmed that the relationship between porosity and mechanical properties of the model GDLs was consistent with those of the actual GDLs. Other physical properties of Model GDL, including thermal and electrical conductivity and permeability, were calculated using Geodict, and it was concluded that GDL sintered at approximately 1000°C would show good performance.

Although the presented relationship between porosity and physical quantity is not surprising, the novelty of paper lies in presenting a scheme for predicting the optimized GDL. I regret to say, there are some sentences where the explanation is insufficient or the basis is not properly shown. It is recommended that the following points are properly addressed before being published.

(1) In line 4 of introduction, Ref[1] is cited to support there are many applications of fuel cells. The placement of this citation seems strange as Ref[1] deals with pyrolysis of printed circuit boards, which does not appear to be relevant to the fuel cell applications.

(2) In line 10 of introduction, Ref[2] is cited to support the the PEM fuel cell uses hydrogen to produce electrical energy. The placement of this citation also seems strange as Ref[2] discuss fabrication of micro porous aluminum.

(3) Although it is written " The porosity of each sample were measured by the image analyzer" in section 2.1, it is unclear how to obtain the experimental value of porsity. Without information on how to obtain the experimental value of porosity and information on accuracy, it is impossible to judge how well the experiment and the model match in Figure 6(b).

(4) According to Section 2.3.2, permeability, K, looks like a tensor. How did you get the scalar value? Also, at what pressure gradient, i.e. which of the direction, did you solve the Stokes-Brinkman equation?

(5) The equation(258, line 4 of section 2.3.4) is not in the form of an equation and is does not make sense.

(6) Some symbols and terms in the equation (262, line 8 of section 2.3.4) is undefined. What is r? What is pc? What is “writing phase contact angle”?

(7) Although it is written "For reaching the appropriate levels of these characteristics, sintering at 1000^circleC has been determined to be the optimal sintering condition" in discussion and conclusion, there basis is not shown. For example, if the appropriate range of characteristics suitable for GDL together with references,  it will be helpful for readers.

Author Response

Reviewer 2.

The authors created a gas diffusion layer (GDL) by tape casting and measured the effects of sintering temperature on mechanical properties and porosity. Model GDLs corresponding to various sintering temperatures were created by piling up grains. It was confirmed that the relationship between porosity and mechanical properties of the model GDLs was consistent with those of the actual GDLs. Other physical properties of Model GDL, including thermal and electrical conductivity and permeability, were calculated using Geodict, and it was concluded that GDL sintered at approximately 1000°C would show good performance.

Although the presented relationship between porosity and physical quantity is not surprising, the novelty of paper lies in presenting a scheme for predicting the optimized GDL. I regret to say, there are some sentences where the explanation is insufficient or the basis is not properly shown. It is recommended that the following points are properly addressed before being published.

Comment 1: In line 4 of introduction, Ref [1] is cited to support there are many applications of fuel cells. The placement of this citation seems strange as Ref [1] deals with pyrolysis of printed circuit boards, which does not appear to be relevant to the fuel cell applications.

Our response to the comment: The authors thank the reviewer for the useful comment. The authors would like to respond that we have rectified the issue by replacing it with the correct reference. Your assistance in improving our paper is greatly appreciated. The new reference can be seen in the given red line.

Modification to the manuscript:

Section “References”:

Jurgen Garche, et al., Application of Fuel Cell Technology: status and Perspectives, The electrochemical society, 2015.

Comment 2: In line 10 of introduction, Ref [2] is cited to support the PEM fuel cell uses hydrogen to produce electrical energy. The placement of this citation also seems strange as Ref [2] discuss fabrication of micro porous aluminum.

Our response to the comment: The authors thank the reviewer for the useful comment. The authors would like to respond that we have checked the Ref [2] and it was not suitable, hence we have replaced that with the correct one. Thanks for helping in improving our paper. The new reference can be seen below in the red line.

Modification to the manuscript:

Section “References”:

Wang, Y., et al., A review of polymer electrolyte membrane fuel cells: Technology, applications, and needs on fundamental research. Applied energy, 2011. 88(4): p. 981-1007.

Comment 3: Although it is written “The porosity of each sample were measured by the image analyzer" in section 2.1, it is unclear how to obtain the experimental value of porosity. Without information on how to obtain the experimental value of porosity and information on accuracy, it is impossible to judge how well the experiment and the model match in Figure 6(b).

Our response to the comment: The authors express gratitude to the reviewer for their valuable comment. In response, we would like to explain our methodology for determining porosity. Firstly, we captured SEM images of the samples at multiple points. Subsequently, using image analyzer software, we obtained 30 readings for each sample and calculated the average porosity to minimize errors. We applied the same approach to determine powder size. The powder size were also measured with the particle analyzer to authenticate our previous reading. The measured porosity and powder size were then used as inputs in the GeoDict modeling. By this way, we tried to mimic the real structure and our models were closely aligned with the experimental results. This reinforces our confidence in the accuracy and reliability of GeoDict. To make this clear the changes in the text can be seen in the given red lines.

Modification to the manuscript:

Section 2.1, paragraph 3, and sentence 3-8:

This sophisticated instrument employs powerful algorithms to analyze the intricate microstructures and extract crucial porosity information. By employing high-resolution imaging and precise image segmentation techniques, we gained a comprehensive understanding of the sample's porosity. This meticulous approach ensures precise and reliable porosity measurements, enabling us to delve deeper into the material's properties and gain valuable insights.

Comment 4: According to Section 2.3.2, permeability, K, looks like a tensor. How did you get the scalar value? Also, at what pressure gradient, i.e. which of the direction, did you solve the Stokes-Brinkman equation?

Our response to the comment:

The authors appreciate the reviewer's valuable comment. In response, we would like to clarify that the background information provided on permeability was intended to highlight the basis for the functionality of FlowDict in GeoDict. GeoDict comprises various modules, each relying on specific equations that are related to experimental work. In each model we do not need to solve any equation we just decide the equation according to the system of our study and then we give some specific inputs that we have to found through experiment like our powder size etc. Considering the porous nature of our material and the limited flow, we selected the Stokes-Brinkman equation as a suitable choice. By utilizing GeoDict and selecting specific modules, we ensured that the equation and modules aligned with our study. We hope this clarification helps to address the reviewer's concerns.

Comment 5: The equation (258, line 4 of section 2.3.4) is not in the form of an equation and is does not make sense.

Our response to the comment: The authors appreciate the reviewer's valuable comment. In response, we acknowledge that the left-hand side of the equation was missing. We have now included the complete equation, which can be found in the indicated red lines. Thank you for bringing this to our attention, and we apologize for any confusion caused.

Modification to the manuscript:

Section 2.3.4, sentence 4-7:

Pore volume distribution = Vcum (di+1) - Vcum (di) / (ln (di+1) – ln (di)).m

Where, d is the pore diameter, v is the volume fraction, Vcum is the cumulative volume fraction and m is the mass.

Comment 6: Some symbols and terms in the equation (262, line 8 of section 2.3.4) is undefined. What is r? What is pc? What is “writing phase contact angle”?

Our response to the comment: The authors appreciate the reviewer's valuable comment. In response, we have addressed this concern by adding the necessary details of the equation, making it clear and comprehensible for the readers. Thank you for bringing this to our attention, and we value your input in improving the clarity of our work.

Modification to the manuscript:

Section 2.3.4, sentence 8-9:

In addition, the bubble point     was computed on the bases of the young Laplace equation

            r =      cos α

Where α is the wetting phase contact angle and the sigma is the surface tension, r is through pore radius [30].

Comment 7: Although it is written "For reaching the appropriate levels of these characteristics, sintering at 1000^circleC has been determined to be the optimal sintering condition" in discussion and conclusion, there basis is not shown. For example, if the appropriate range of characteristics suitable for GDL together with references, it will be helpful for readers.

Our response to the comment: The authors express their gratitude to the reviewer for their valuable comment. In response, we would like to highlight that the Gas Diffusion Layer (GDL) was optimized by considering two important properties: permeability and mechanical strength. To ensure clarity for the reader, we have added explanatory text, which is marked by red lines below.

Modification to the manuscript:

Section 3.3, paragraph 2-4:

The optimization of the sintering temperature for GDL fabrication in PEM electrolyzers is crucial, taking into account key factors such as permeability and mechanical properties. Extensive studies have shown that a permeability above 10-12 m2 is desirable for achieving high efficiency in GDLs. Unfortunately, the sample sintered at 1100°C and above results in lower permeability values that do not meet this threshold, rendering them unsuitable for efficient applications. [36-41].

On the other hand, lower sintering temperatures like 800°C and 900°C may yield higher permeability, but they lack the necessary mechanical strength required for handling and testing in real stacks. These samples tend to be fragile and prone to breakage, making them impractical for implementation.

By selecting a sintering temperature of 1000°C, a delicate balance is achieved between mechanical strength and permeability. Samples sintered at this temperature exhibit sufficient strength to withstand handling and fluid pressure, while still maintaining a relatively high permeability compared to high temperature (above 1000°C) sintered samples.  Overall, the optimization of the sintering temperature at 1000°C ensures that the GDL exhibits both desirable mechanical strength and adequate permeability for efficient operation in PEM electrolyzers.

Reviewer 3 Report

This paper aims to investigate and developed a single-layer Ti64 GDLvia tape casting. The tape was cast and sintered at different temperatures that ranges from8000C-1400 C. Then the characterization was done and nanoindentation was used to findthe mechanical properties. On the basis of GeoDict simulations further prediction and optimization of sintering w.r.t porosity was done. However, the paper contains many descriptive errors and lacks scientific analysis and has not yet met the standards for publication. The specific problems are as follows.

1. In the Introduction section, regarding the simulation for the optimization of GDL material properties, the authors directly introduce the GeoDict software tool, do other simulation methods exist? It is recommended that the authors provide an overview of the commonly used numerical methods and compare them. In addition, the simulation work for GeoDict presents the results of only two researchers, and the review is not comprehensive enough.

2. In section 2.2.1, ‘a distribution process was carried out and 140 iterations were selected for a more uniform distribution’. How is the number of iterations of 140 determined and what is the convergence condition, please specify.

3. Incorrect captions for Figure 7 (c) and Figure 7 (d).

4. The captions of Figure 8 (a) and (b) are wrong.

5. Figure 8 gives the relationship between porosity and thermal conductivity, electrical conductivity and resistivity, but the author only describes in general terms that porosity has a significant effect, without describing the phenomenon shown in the graph and explaining in detail the causes of this phenomenon. Please add a description of the graph and an analysis of the causes.

6. The description of Figure 9 seems to be inaccurate. The figure shows that the larger the porosity the lower the number of pores, however the author describes that the porosity is quite high, and the pores are numerous with large diameters. Please redescribe.

7. The manuscript mentions in section 3.3 that the average velocity decreases with decreasing porosity, which is not the result of the simulation data response, and there is no graph about the relationship between average velocity and porosity in the manuscript, please describe it scientifically.

8. In section 3.3, it is concluded that GDL has the best performance at 1000°C, but there is no data analysis, so it is not clear how this conclusion is obtained.

Author Response

Reviewer 3.

This paper aims to investigate and developed a single-layer Ti64 GDL via tape casting. The tape was cast and sintered at different temperatures that ranges from8000C-1400 C. Then the characterization was done and nanoindentation was used to find the mechanical properties. On the basis of GeoDict simulations further prediction and optimization of sintering w.r.t porosity was done. However, the paper contains many descriptive errors and lacks scientific analysis and has not yet met the standards for publication. The specific problems are as follows.

Comment 1: In the Introduction section, regarding the simulation for the optimization of GDL material properties, the authors directly introduce the GeoDict software tool, do other simulation methods exist? It is recommended that the authors provide an overview of the commonly used numerical methods and compare them. In addition, the simulation work for GeoDict presents the results of only two researchers, and the review is not comprehensive enough.

Our response to the comment: The authors appreciate the reviewer's helpful comment. In response, we have included a brief description of other simulation methods and provided reasons for choosing GeoDict as our preferred software. Furthermore, we have added additional references to further validate the use of GeoDict as shown in the 2nd paragraph of red lines. These references also include a comparison with two other software options (StarCCM+ and COMSOL). The additions can be found in the highlighted paragraphs. Thank you for pointing out the need for this information, and we believe these updates enhance the credibility and robustness of our study. The addition can be seen in the below red lines.

Modification to the manuscript:

Section 1.2, paragraph 1, sentence 1-10 and paragraph 3, sentences 7-17:

Currently, there is limited simulated research on Titanium GDL manufactured by powder metallurgy. However, software prediction is expected to play a crucial role in the future for GDL design and property optimization. Computational fluid dynamics (CFD) provides insights into macroscopic fluid behavior, while finite element analysis (FEA) focuses only on mechanical aspects. Molecular dynamics (MD) simulations has specialty in offering atomistic-level insights.  Geodict possesses distinctive functionalities that enable the generation of realistic 3D micro-sized geometries, simulation of fluid flow and transport phenomena, and analysis of the mechanical behavior of GDL materials. These exclusive capabilities make Geodict an ideal tool for investigating complex transport phenomena and optimizing the performance of GDLs in PEMFCs.

In a study conducted by Dennis Hoch et al., a comparison was made between GeoDict and StarCCM+ using the experimental results. The findings revealed that while StarCCM+ exhibited almost 1% higher accuracy, but its simulation time was significantly longer and less convenient compared to GeoDict [20]. According to M. Amin et al. [21], compared the results of GEODICT and COMSOL with the experimental results and it was revealed that GeoDict demonstrated a high level of accuracy in its predictions. H. Bai et al. utilized FilterDict to simulate the effective diffusion coefficient and filtration performance in relation to particle diameter. The results showed a deviation of approximately 0.6% from the experimental data, thus confirming the remarkable accuracy of GeoDict [22].

Comment 2: In section 2.2.1, ‘a distribution process was carried out and 140 iterations were selected for a more uniform distribution’. How is the number of iterations of 140 determined and what is the convergence condition, please specify.

Our response to the comment: The authors thank the reviewer for the useful comment. The authors would like to respond that in GeoDict, increasing the iteration leads to a more uniform distribution, but it also increases the computational time. To optimize this balance, I conducted a series of iterations, starting from 20 and incrementing by 20. Upon reaching 140 iterations, I observed the most uniform distribution. Further increasing the iteration did not provide significant benefits, but it significantly increased the time required. Therefore, I determined that 140 iterations represented the optimal condition, striking a balance between achieving a uniform distribution and minimizing computational time.

Comment 3: Incorrect captions for Figure 7 (c) and Figure 7 (d).

Our response to the comment: The authors appreciate the valuable feedback provided by the reviewer. In response, we have made the necessary corrections to the captions of Figure 7c and 7d, as indicated below.

Modification to the manuscript:

Section 3.2, Figure 7:

Fig. 7. (a) Relation of porosity with the poison ratio. (b) Relation of Permeability and average velocity with porosity, (c) relation of tortuosity with porosity, and (d) porosity and bubble point relationship

Comment 4: The captions of Figure 8 (a) and (b) are wrong.

Our response to the comment: The authors thank the reviewer for the useful comment. The authors acknowledge the reviewer's comment. We have addressed the concern by making the necessary corrections to the captions of Figure 8a and 8b, as presented below.

Modification to the manuscript:

Section 3.2, Figure 8:

Fig. 8. (a) Porosity and thermal conductivity, and (b) Relation of porosity with electrical conductivity and electrical resistivity.

Comment 5:  Figure 8 gives the relationship between porosity and thermal conductivity, electrical conductivity and resistivity, but the author only describes in general terms that porosity has a significant effect, without describing the phenomenon shown in the graph and explaining in detail the causes of this phenomenon. Please add a description of the graph and an analysis of the causes.

Our response to the comment: The authors appreciate the valuable feedback provided by the reviewer. In response, we have included additional details in the manuscript to enhance the reader's understanding of the relationship between porosity and thermal/electrical conductivity. We have also highlighted the reasons behind the observed changes in conductivity with increasing porosity. The specific additions made to address these points can be identified in the below paragraphs

Modification to the manuscript:

Section 3.2, paragraph 3, sentences 5-19:

With an increase in porosity, the electrical and thermal properties exhibit a discernible decline, highlighting a crucial relationship between these properties and the porosity of the material. This phenomenon can be attributed to several underlying reasons. Firstly, as porosity increases, the material's effective cross-sectional area decreases. This reduction in the available pathway for electrical and thermal conduction leads to a decrease in conductivity and thermal diffusivity. The presence of voids and air-filled spaces within the material interrupts the flow of electrons and heat, impeding their efficient transfer. Secondly, the interconnected voids and pores within the porous structure create additional interfaces, introducing resistance to the movement of charge carriers and thermal energy. These interfaces act as barriers, hindering the smooth flow of electrons and heat through the material. Consequently, electrical and thermal conductivities are reduced. It is worth noting that the extent of the decrease in electrical and thermal properties depends on various factors, including the porosity volume fraction, pore size, and distribution. Materials with higher porosity and larger pore sizes tend to exhibit more pronounced decreases in these properties.

Comment 6: The description of Figure 9 seems to be inaccurate. The figure shows that the larger the porosity the lower the number of pores, however the author describes that the porosity is quite high, and the pores are numerous with large diameters. Please re describe.

Our response to the comment: The authors express their gratitude to the reviewer for their valuable comment. In response, we have carefully revised the description based on the suggestions provided. The specific changes made can be observed in the highlighted sections below.

Modification to the manuscript:

Section 3.2, paragraph 5, and sentence 1-8:

During the initial stages of the sintering process, the material exhibits a higher porosity, characterized by a low volume of grains and relatively large pore diameters, particularly open pores, as depicted in Figure 9 (a, b, c, and d). As the temperature rises, the number of pores increases while their diameters decrease. Simultaneously, the number of grains decreases, and their volumes diminish with the escalating temperature. At higher porosity levels and larger pore diameters, a wider pathway is established for fluid flow, resulting in enhanced permeability and average velocity. These conditions facilitate a smoother and more defined movement of the fluid within the material.

Comment 7: The manuscript mentions in section 3.3 that the average velocity decreases with decreasing porosity, which is not the result of the simulation data response, and there is no graph about the relationship between average velocity and porosity in the manuscript, please describe it scientifically.

Our response to the comment: The authors appreciate the reviewer's helpful comment. In response, we have included the missing caption for the figure illustrating the relationship between porosity and average velocity. Additionally, we have made further changes in the text to provide clearer explanations for the reader. These modifications can be observed in the highlighted paragraph below.

Modification to the manuscript:

Section 3.2, paragraph 2, and sentence 6-8:

The decrease in fluid velocity with decreasing porosity is a result of reduced available flow pathways, increased resistance to flow, enhanced fluid-solid interactions, and higher tortuosity within the porous material.

Figure 7. (a) Relation of porosity with the poison ratio. (b) Relation of Permeability and average velocity with porosity, (c) relation of tortuosity with porosity, and (d) porosity and bubble point relationship

Comment 8: In section 3.3, it is concluded that GDL has the best performance at 1000°C, but there is no data analysis, so it is not clear how this conclusion is obtained.

Our response to the comment: The authors thank the reviewer for the useful comment. The authors would like to respond that

Modification to the manuscript:

Section 3.3, paragraph 2-4:

The optimization of the sintering temperature for GDL fabrication in PEM electrolyzers is crucial, taking into account key factors such as permeability and mechanical properties. Extensive studies have shown that a permeability above 10-12 m2 is desirable for achieving high efficiency in GDLs. Unfortunately, the sample sintered at 1100°C and above results in lower permeability values that do not meet this threshold, rendering them unsuitable for efficient applications. [36-41].

On the other hand, lower sintering temperatures like 800°C and 900°C may yield higher permeability, but they lack the necessary mechanical strength required for handling and testing in real stacks. These samples tend to be fragile and prone to breakage, making them impractical for implementation.

By selecting a sintering temperature of 1000°C, a delicate balance is achieved between mechanical strength and permeability. Samples sintered at this temperature exhibit sufficient strength to withstand handling and fluid pressure, while still maintaining a relatively high permeability compared to high temperature (above 1000°C) sintered samples.  Overall, the optimization of the sintering temperature at 1000°C ensures that the GDL exhibits both desirable mechanical strength and adequate permeability for efficient operation in PEM electrolyzers.

Round 2

Reviewer 3 Report

The author has carefully revised and improved the manuscript and suggestions raised by the reviewers, resulting in a significant improvement in the quality of the manuscript. I recommend accept by this version.